# Autologous Platelet-Rich Growth Factor Reduces M1 Macrophages and Modulates Inflammatory Microenvironments to Promote Sciatic Nerve Regeneration

**DOI:** 10.3390/biomedicines10081991

**Published:** 2022-08-17

**Authors:** Anjali Yadav, Thamil Selvee Ramasamy, Sheng-Che Lin, Szu-Han Chen, Jean Lu, Ya-Hsin Liu, Fu-I Lu, Yuan-Yu Hsueh, Shau-Ping Lin, Chia-Ching Wu

**Affiliations:** 1Department of Cell Biology and Anatomy, College of Medicine, National Cheng Kung University, Tainan 701, Taiwan; 2International Center for Wound Repair and Regeneration, National Cheng Kung University, Tainan 701, Taiwan; 3Stem Cell Biology Laboratory, Department of Molecular Medicine, Faculty of Medicine, University of Malaya, Kuala Lumpur 50603, Malaysia; 4Division of Plastic and Reconstructive Surgery, Tainan Municipal An-Nan Hospital-China Medical University, Tainan 709, Taiwan; 5Division of Plastic and Reconstructive Surgery, Department of Surgery, National Cheng Kung University Hospital, Tainan 701, Taiwan; 6Institute of Clinical Medicine, College of Medicine, National Cheng Kung University, Tainan 701, Taiwan; 7Genomics Research Center, Academia Sinica, Taipei 115, Taiwan; 8Department of Life Sciences, College of Bioscience and Biotechnology, National Cheng Kung University, Tainan 701, Taiwan; 9Department of Biotechnology and Bioindustry Sciences, College of Bioscience and Biotechnology, National Cheng Kung University, Tainan 701, Taiwan; 10Institute of Biotechnology, College of Bio-Resources and Agriculture, National Taiwan University, Taipei 106, Taiwan; 11Department of Biomedical Engineering, National Cheng Kung University, Tainan 701, Taiwan

**Keywords:** platelet-rich growth factors, sciatic nerve injury, peripheral nerve regeneration, macrophage polarization

## Abstract

The failure of peripheral nerve regeneration is often associated with the inability to generate a permissive molecular and cellular microenvironment for nerve repair. Autologous therapies, such as platelet-rich plasma (PRP) or its derivative platelet-rich growth factors (PRGF), may improve peripheral nerve regeneration via unknown mechanistic roles and actions in macrophage polarization. In the current study, we hypothesize that excessive and prolonged inflammation might result in the failure of pro-inflammatory M1 macrophage transit to anti-inflammatory M2 macrophages in large nerve defects. PRGF was used in vitro at the time the unpolarized macrophages (M0) macrophages were induced to M1 macrophages to observe if PRGF altered the secretion of cytokines and resulted in a phenotypic change. PRGF was also employed in the nerve conduit of a rat sciatic nerve transection model to identify alterations in macrophages that might influence excessive inflammation and nerve regeneration. PRGF administration reduced the mRNA expression of tumor necrosis factor-α (TNFα), interleukin-1β (IL-1β), and IL-6 in M0 macrophages. Increased CD206 substantiated the shift of pro-inflammatory cytokines to the M2 regenerative macrophage. Administration of PRGF in the nerve conduit after rat sciatic nerve transection promoted nerve regeneration by improving nerve gross morphology and its targeted gastrocnemius muscle mass. The regenerative markers were increased for regrown axons (protein gene product, PGP9.5), Schwann cells (S100β), and myelin basic protein (MBP) after 6 weeks of injury. The decreased expression of TNFα, IL-1β, IL-6, and CD68^+^ M1 macrophages indicated that the inflammatory microenvironments were reduced in the PRGF-treated nerve tissue. The increase in RECA-positive cells suggested the PRGF also promoted angiogenesis during nerve regeneration. Taken together, these results indicate the potential role and clinical implication of autologous PRGF in regulating inflammatory microenvironments via macrophage polarization after nerve transection.

## 1. Introduction

The success of peripheral nerve regeneration relies on both neuronal and non-neuronal cells, such as macrophages [1,2]. Damaged Schwann cells (SCs) and axons secrete chemo-attractive factors, TNFα and IL-1β, for the recruitment of macrophages [3]. Macrophages recruited at the injury site, in turn, secrete cytokines and growth factors to generate a favorable microenvironment for repair by removing inhibitory components, such as axonal and myelin debris [4,5]. However, the switch of macrophages from the pro-inflammatory (M1) to the anti-inflammatory (M2) phenotype is essential for the regulation of initial events of inflammation [6]. The functional state of the macrophage is affected by the local milieu and M2 macrophages contribute to the repair process by producing anti-inflammatory cytokines and growth factors [7,8]. More than the extent of availability of macrophages at the injury site, their phenotypes influence nerve regeneration [6]. The slow transition from M1 to M2 macrophage [9] and prolonged inflammation [10] are amongst the factors that affect axonal regrowth and eventually obstruct nerve regeneration. Sluggish axonal regrowth often leads to failure of the nerve to reinnervate the target tissue and restore function [11]. The chronic constriction injury model has also shown that sustained levels of pro-inflammatory cytokines (TNFα and IL-1β) and persistent infiltration of macrophages result in neuropathy and muscle weakness [12]. As a result, immune modulation has emerged as a therapeutic target to improve regeneration by regulating inflammation [13,14].

Growth factors, such as nerve growth factor (NGF) and fibroblast growth factors (FGF) are present in the nerves in a physiological state and create a regenerative microenvironment at the time of injury for axonal outgrowth [15]. However, endogenous levels of growth factors are insufficient to support axonal regeneration and remyelination [16]. Administration of high doses of growth factors with biomaterials has been used for the repair of a peripheral nerve injury [17,18]. The filling of silicon conduit with NGF improved nerve regeneration, reduced the secretion of IL-1 cytokine, and increased the paracrine production of FGFs and platelet-derived growth factors (PDGF) [19]. Similarly, transplantation of FGF-9-induced neural lineage-like cells (NLCs) in the nerve conduit after sciatic nerve transection injury improved the axonal regrowth, remyelination, and reinnervation of the targeted muscle [20]. However, the application of the growth factors at the injury site has its own challenges, such as diffusion of growth factors from the site, short half-life, and proteolysis of growth factors [21].

Researchers have started utilizing autologous therapies, such as platelet-rich plasma (PRP) and its variations, in the practice of regenerative medicine [22]. PRP is an autologous biological product enriched with a high concentration of growth factors and has been used to promote wound healing and tissue regeneration. It has been suggested to have a platelet concentration three-folds higher in PRP and its derivatives than in whole blood to obtain the regenerative outcomes [23,24]. PRP prevents the production of a high level of pro-inflammatory TNFα, nitric oxide, and cyclooxygenase in a mouse model of Parkinson’s disease [25]. However, the presence of white blood cells (WBCs) in addition to platelets in the PRP causes a release of pro-inflammatory cytokines at the injury site producing an augmented inflammatory response [26]. Similarly, the pro-inflammatory nature of PRP is found to be detrimental to cell growth [27]. Therefore, researchers have developed a PRP derivative known as platelet-rich growth factor (PRGF), through single slow centrifugation of whole blood which prevents the formation of a buffy coat that contains WBCs along with platelets [28]. PRGF is beneficial in reducing the production of inflammatory cytokines, such as TNFα and IL-1β for the management of atopic dermatitis condition [29].

PRP has been shown to improve motor and sensory functions after nerve injury or neuropathies [30]. Similarly, the administration of PRP and its derivative platelet-rich fibrin (PRF) improved functional recovery in a rat model of a sciatic nerve injury [31]. PRGF also increases nerve regeneration in the early periods of nerve healing [32]. Studies have been published regarding the use of PRGF and its positive effects on nerve regeneration, but no one has reported its effect on the regulation of inflammation and macrophages in a nerve. Furthermore, the mode of action of PRGF on the macrophages is yet to be elucidated. Although PRP and other platelet-derived products have demonstrated effects on macrophage polarization [33,34,35,36], the effects of PRGF on chronic inflammation and macrophage polarization during nerve regeneration have not been examined. In this study, we examined the dosage of PRGF that would affect the expression and secretory profile of pro-inflammatory cytokines in M1 macrophages and SCs in vitro. The administration of autologous PRGF (a pool of growth factors from blood plasma without WBC) into nerve conduits during nerve surgery and investigation into how to modulate an inflammatory microenvironment were carried out in a rat sciatic nerve transection model. Our results reveal the active roles of PRGF on macrophages, SCs, and regrowth axons, which may provide valuable information for the development of therapeutic strategies for nerve regeneration.

## 2. Materials and Methods

### 2.1. PRGF Preparation for In Vitro and In Vivo Usage

A group of 8-week-old male Sprague Dawley (SD) rats weighing 300–350 g (BioLASCO Taiwan Co Ltd., Taipei, Taiwan) was kept at the animal center with approval documents from the Institutional Animal Care and Use Committee (IACUC-105224) at the National Cheng Kung University (NCKU). After the rats were anesthetized using isoflurane (Panion and BF Biotech Inc., Taipei, Taiwan), blood was collected from the jugular vein. The rats were kept under observation until they opened their eyes, before being taken back to the cages. The maximum amount of blood collected at each time from one rat was no more than 3 mL. The same rats were not used again for 7 days to collect blood to ensure their good health. The procedure did not harm the animals in any way. After blood collection, the blood was diluted with 3.8% sodium citrate (3646-01, JT Baker, Avantor, Pennsylvania, USA) in a volume ratio of 1:20 in an Eppendorf and was gently inverted a few times to prevent coagulation. The tubes were spun at 1900 rpm for 8 min at 27 °C. The blood plasma was separated from the red blood cells (RBCs) and the fuzzy layer of WBCs. The lower plasma (fraction 2) was transferred into a new tube containing 10% CaCl_2_ to activate platelets for gel formation by incubation at 37 °C for 1 h. Further centrifugation of the tubes at 1000× *g* for 20 min at 4 °C resulted in a distinct separation of liquid which was transferred into new tubes and defined as PRGF. For in vivo experiments, the autologous PRGF was prepared freshly by collecting blood from each rat before the surgery.

### 2.2. Culture and Treatment of THP-1 Cells and RT4 SCs

To study macrophage modulation, a human monocytic cell line (THP-1, Bioresource Collection and Research Center, Taiwan) was used according to a previously described protocol [9,37]. Experiments were conducted at a cellular concentration of 1.5 × 10^5^ cells/mL. Briefly, after the THP-1 monocytes were differentiated into M0 macrophages by incubating with 150 nM phorbol 12-myristate 13-acetate (PMA) for 24 h, the cells were washed with phosphate-buffered saline (PBS) to eliminate the traces of PMA and allowed to grow in RPMI media containing 10% fetal bovine serum (FBS), 100 U/mL penicillin (Gibco, USA), and 100 µg/mL streptomycin (Gibco, Thermo Fisher Scientific, Waltham, MA, USA) for 24 h (M0 medium). The following day M0 macrophages were treated with M1 medium which contained the lipopolysaccharide (LPS) and interferon γ (IFNγ) to induce the M1 phenotype for 24 h. A different percentage of PRGF was mixed into the M0 or M1 medium to study the influence of PRGF on macrophage polarization. Without notice, the 10% PRGF was used to investigate the treatment outcomes.

RT4-SCs (RT4, ATCC number CRL-2768), which is a rat Schwann Cell line, were cultured in Dulbecco’s Modified Eagle Medium (DMEM) containing 10% HyClone FBS (Cytiva, Logan, UT, USA), 100 U/mL penicillin (Gibco, Thermo Fisher Scientific, Waltham, MA, USA), and 100 µg/mL streptomycin (Gibco, Thermo Fisher Scientific, Waltham, MA, USA). The cells were maintained at 37 °C with 5% CO_2_ A total of 3 × 10^5^ cells were seeded in a 6 cm culture dish and grown for 24 h until they reached 80% confluency. The cells were treated for 3 h with 1 µg/mL of lipopolysaccharide (LPS) to induce inflammation [10]. The anti-inflammatory effect of PRGF was tested by mixing 10% PRGF with an LPS-containing medium. The media from the treated cells was collected and used to perform ELISA.

### 2.3. Gene Expression Measurement Using Quantitative Real-Time Polymerase Chain Reaction (qRT-PCR)

The cells after the treatment period were washed with ice-cold PBS, and 1 mL of Trizol was added into each 6 cm dish for RNA extraction. The RNA was extracted manually using the chloroform–isopropanol method relying on an aqueous and organic phase separation [38]. After determining the concentration and quality of RNA using spectrophotometry (a Nabi-UV/Vis Nano Spectrometer, MicroDigital Co., Ltd., Seoul, Korea), 5 μg of RNA was used for reverse transcription to complementary DNA (cDNA) as per the manufacturer’s protocol (SuperScript III Cells Direct cDNA Synthesis System) [39]. The cDNA was diluted 5x using MQ water and further used for qPCR to examine the mRNA expression of inflammatory (TNFα, IL-1β, IL-6, and IL-10) and macrophage (CD86 and CD206) genes. The sequence of the forward and reverse primers is listed in Table 1. The qRT-PCR resulted in Ct values. We used ΔΔCt to obtain the normalized relative expression. Briefly at first, we normalized the expression values (ΔCt for each condition; difference between Ct value of gene of interest and housekeeping gene). In the next step, we calculated the difference between ΔΔCt from the control (M0) and experimental groups. The last step was to calculate the changes in gene expression levels by 2^(ΔΔCt).

### 2.4. Enzyme-Linked Immunosorbent Assay (ELISA) for TNFα Secretion

The cell-cultured media from different experimental groups were collected and centrifuged at 200 rcf for 5 min to remove the dead cells. To measure the secretion of TNFα, we followed the protocol illustrated in the human TNFα ELISA Kit (430207, BioLegend Inc, San Diego, CA, USA). In short, all the samples and solutions were brought to RT before starting the experiment and the wells in the strip were thoroughly washed in between each step using 1X wash buffer. The standard dilutions and samples were added to wells and incubated for 2 h at RT with shaking. Then, a human TNFα detection antibody was added to each well and incubated for 1 h at RT while shaking. HRP-D solution was added after the wash and kept for 30 min. At the end, the substrate solution F was added to each well until the blue color developed (~2 min); the blue color intensity was proportional to the TNFα levels in the sample. The development of color was stopped using a stop solution, and the absorbance was read at 450 nm and 570 nm using an ELISA reader (μQuant, Bio-Tek Instruments, Inc., Winooski, VT, USA). A calculation to determine the concentration of TNFα expression in pg/mL was carried out using the standard curve obtained in each individual experiment.

### 2.5. The Animal Model for Sciatic Nerve Regeneration

A group of 8-week-old male SD rats weighing 300–350 g were kept at the animal center of NCKU before and after surgery. The animals were housed at 21 ± 0.5 °C in cages, with free access to food and water. The procedure for surgery was approved after review by the Institutional Animal Care and Use Committee (IACUC-105224) at NCKU. The animals were classified into three groups: sham, PBS, and PRGF (a total of 8 rats; 2 for sham, 3 for each PBS and PRGF group). Firstly, the rats were anesthetized by an intraperitoneal injection of 25 mg/kg Tiletamine and 25 mg/kg Zolazepam (Zoletil, Virbac Lab, Carros, France). The surgery followed the same method as described in our previous study [10]. Briefly, for sham group animals, the left sciatic nerve was exposed without making any transection to it while the PBS- and PRGF-group animals received a transection in the left nerve. The two transected nerve ends were sutured using a silicon conduit (0.15 mm inner diameter, 1.2 cm in length) (Versilic® 760110, Saint-Gobain, Courbevoie, France) to create a 1.2 cm gap with the administration of either PBS or 10% PRGF in the conduit. The conduit contained a 100 μL volume of either PBS or 10% PRGF in total and was administered only once at the time of surgery. The muscle and the skin were sutured back using No. 4-0 nylon suture (Ethicon US, Bridgewater, NJ, USA). The rats were injected with 1 mg/kg ketorolac to relieve pain for 5 days post-surgery and covered with a neck collar to avoid auto-mutilation for 10 days. Post-6 weeks of surgery the animals were euthanized to harvest left and right gastrocnemius muscles along with the left nerve for their further immunohistochemical analysis.

### 2.6. Paraffin Block Preparation and Immunohistochemistry (IHC)

The middle region of the nerves harvested post-6 weeks injury was fixed using 4% paraformaldehyde overnight at 4 °C. The fixed nerves were dehydrated for 1 h each in a series of ethanol solutions: 70%, 75%, 85%, 95%, and 100%. To clear the penetrated ethanol from the tissues they were kept in xylene washes for 30 min, five times. Following this, the xylene was removed from the tissues by immersing them in paraffin at (5 times, 30 min each) at 65 °C. Finally, the tissues are embedded in clear paraffin wax to make blocks and stored at room temperature (RT) but transferred to −20 °C for 1 h before sectioning. On the aminopropyltriethoxysilane-coated slides (APS, Matsunami Glass Ind., Ltd., Kishiwda, Japan), the 10 µm transverse sections were cut using a microtome were collected and dried at 40 °C to remove the water from the tissues. The slides were stored at RT until use. For performing IHC, the nerve sections on the slides were rehydrated by immersion in xylene twice and once in each ethanol solution (100%, 95%, 85%, and 75%). The antigens were unmasked by incubating in boiling antigen retrieval buffer (Thermo Fisher Scientific, Waltham, MA, USA) for 20 min. The slides were then blocked and processed further as per the instructions in the IHC kit (ab232466, Abcam, Cambridge, MA, USA). The IHC on the nerves was performed for the following primary antibodies: PGP9.5 (1:100, ab109261, Abcam, Cambridge, MA, USA), S100β (1:1000, ab52642, Abcam, Cambridge, MA, USA), myelin basic protein (MBP, 1:500, 78896T, Cell Signaling Technology, Danvers, MA, USA), TNF-α (1:1000, ab1793, Abcam, Cambridge, MA, USA), IL-1β (1:1000, ab9722, Abcam, Cambridge, MA, USA), IL-6 (1:1000, ab9324, Abcam, Cambridge, MA, USA), and CD86 (1:500, ab53004, Abcam, Cambridge, MA, USA), by overnight incubation at 4 °C. After obtaining the DAB signal, staining with hematoxylin, and mounting, the IHC was captured using Magnafire software with an Olympus microscope at 40× magnification.

The analysis was performed by acquiring three images from three random visual fields in three different sections for each nerve tissue harvested from different treatments. The photomicrographs were captured at 40× magnification by using Magnafire software with an Olympus microscope. The three visual fields at 40× magnification covered 0.28 mm^2^ of the area of each section. For the IHC staining of S100β, PGP9.5, MBP, CD68, and RECA, the number of positive cells in 0.28 mm^2^ was manually counted and extrapolated to the number of positive cells/mm^2^. Since the signal of TNFα, IL-1β, and IL-6 was present even outside the cells, we referred to the IHC image quantification protocol described in our previous article [10]. In brief, a threshold value was set to remove the background signal after the deconvolution of the IHC images, followed by the quantification of the DAB signal within the image. The hematoxylin signal was also quantified by using a similar process to the DAB signal. Both the DAB and hematoxylin intensity was then normalized against the cell nuclear intensity. The average intensity of the DAB signal acquiring three images from three random visual fields in one section in the IHC images was calculated and the mean value of three visual fields was chosen to represent the target protein expression.

### 2.7. Statistical Analysis

Each experiment was carried out ‘n’ times on different days using the same protocol to analyze its significance. The n for each experiment is mentioned in the figure legends. The statistical analysis was evaluated using the distinct kinds of tests in GraphPad. A paired two-tailed Student’s *t*-test was performed when the two groups were a part of the study. A result was deemed to be significant if the *p*-value < 0.05.

## 3. Results

### 3.1. PRGF Attenuates the Pro-Inflammatory Cytokines during M1 Macrophage Induction

Activated M1 macrophages increased the production of TNFα. To test the dosage responses of PRGF during M1 induction, various doses of PRGF (1, 5, and 10%) were mixed with M0 culture medium or M1 induction medium in monocyte-derived macrophages (M0) cells. The secretion of TNFα and IL-10 were measured by examining the supernatant after 24 h of treatments using ELISA. The TNFα secretion was significantly increased in the M1-induced macrophages after LPS and IFNγ stimuli (M1 induction medium) (Figure 1A). Application of PRGF gradually reduced the TNFα secretion in a dosage-dependent manner, especially for the 5% and 10% PRGF (Figure 1A). The mixture of 10% PRGF in M1 induction medium significantly increased the IL-10 secretion with respect to both M0 and M1 macrophages (Figure 1B). Hence, we selected 10% PRGF as the PRGF treatment percentage for the following in vitro experiments. M0 macrophages polarized to the pro-inflammatory M1 phenotype with LPS and IFNγ also released cytokines, such as TNFα, IL-1β, and IL-6. The qPCR results showed decreased mRNA expression levels of prototypical M1 cytokines; TNFα, IL-1β, and IL-6 by the treatment of M0 macrophages activated to M1 macrophages with PRGF for 24 h (Figure 1C). These results indicated the potential of PRGF in modulating the inflammatory cytokine profile of M1 macrophages.

### 3.2. Regenerative Phenotypic Changes of Macrophages and SCs with PRGF Treatments

Since the secreted cytokines may correlate with distinct cell morphology and phenotypic changes, we further investigated macrophage polarization and glia activation in response to PRGF. The light microscopy phase images showed a morphological change of a polarized and elongated M1 phenotype after being subjected to the M1 induction medium for 24 h (Figure 2A). The application of 10% PRGF to the M1 induction medium inhibited the M1 phenotype and kept the macrophage in a round shape. The non-polarized macrophages also switched to M2 macrophages to serve the pro-healing functions. We further tested if the PRGF treatment could promote polarization toward M2 macrophages and characterized its expression on the mannose receptor (CD206) marker. The qPCR results depicted an increase in CD206 expression levels after incubation with M1-induced macrophages and PRGF in comparison to only M1-induction or M0 culture media (Figure 2B). Likewise, CD86 expression levels also increased after PRGF treatment (Figure 2B). This increase was probably due to a subset of the population not transiting from an M1 macrophage to an M2 macrophage.

The increases in TNFα, IL-1β, and IL-6 in SCs confirmed the glia cell activation toward the pro-inflammatory stage (Figure 2C). The administration of 10% PRGF significantly reduced the mRNA expression levels of inflammatory cytokines IL-1β and IL-6 in the LPS-induced RT4 SCs (Figure 2C). Taken together, these results suggested that PRGF may promote regenerative phenotypic changes in both macrophages and SCs.

### 3.3. PRGF Alleviates Chronic Inflammation through Suppression of the M1 Macrophage during Nerve Regeneration

The therapeutic effects of PRGF were further investigated on nerve tissue regeneration by using the sciatic nerve transection model and by administering autologous PRGF into a silicon conduit in rats. After 6 weeks post-injury, the nerve tissues were harvested and showed the PRGF-treated nerve had thicker gross morphology of regrown nerve tissue (middle section) compared to the almost-failed regeneration in the saline-treated (negative control) nerve conduit (Figure 3A). To confirm our in vitro finding of PRGF in anti-inflammatory and M1 macrophage inhibition, we measured the tissue inflammation levels by performing IHC staining on the transverse sections of the middle region of the nerve harvested at 6 weeks post-injury. The pro-inflammatory cytokine markers for TNFα, IL-1β, and IL-6 were reduced in the PRGF-treated nerve compared to the saline control group (Figure 3B). The number of pro-inflammatory M1 macrophages was reduced in PRGF-treated groups as indicated by the decrease in CD-68 positive cells (Figure 3C). These results suggested that PRGF administration might provide a supportive microenvironment by inflammatory modulation for nerve regeneration.

### 3.4. PRGF Fosters Nerve Regeneration and Reinnervation through an Increase in SCs Number, Regrown Axons, and Remyelination

Although PRGF improved nerve healing by restoration of the function in peripheral nerves [32], detailed regenerative assessments on axonal regrowth and remyelination are yet to be delineated. The IHC staining on transverse sections of the middle region of the nerve was carried out for various markers, such as PGP9.5 (axonal regrowth), S100β (the number of SCs), and MBP (myelinated SCs). An increase in PGP9.5-positive axons at the core region of the nerve fiber was seen in the PRGF-treated group, indicating better axonal regrowth by PRGF administration (Figure 4A). Meanwhile, a marked change in the number of SCs was observed in the PRGF-treated nerve compared to the saline group, as reflected by more positively stained S100β cells (brown color) with myelin sheath morphology at 6 weeks post-injury (Figure 4A). The myelination of regrown axons was further demonstrated by the increased number of MBP positively stained cells after PRGF treatment (Figure 4A). In addition to the hint of promoting the M2 macrophage from our in vitro results, we also assessed the possibility of releasing other regenerative factors from the M2d subtype macrophage to produce angiogenic factor VEGF and promote angiogenesis. The IHC stain for the endothelial cell marker RECA-1 showed an increase in the number of RECA-positive cells in the PRGF group compared to the PBS control group (Figure 4B). Thus, both remyelination and angiogenesis were observed in the regenerated nerve after PRGF treatment.

The toluidine-blue staining confirmed improvements in the myelin sheath formation in the PRGF-treated rats (Figure 5A). The target innervation muscles of the sciatic nerve were bilaterally harvested on the gastrocnemius muscle of two legs to compare the re-innervation of an injured nerve from the normal side. The left gastrocnemius muscle associated with the transected nerve showed better morphology and retention of muscle mass in the PRGF-treated group in comparison to saline indicating the prevention of muscle atrophy by PRGF administration at the injury site (Figure 5B). The quantification of the targeted gastrocnemius muscle was measured by the relative gastrocnemius muscle weight (RGMW) of the left (transected side) and right (un-transected side) muscles. The PRGF-treated rat showed a significant change in the RGMW percentage compared to the saline group (Figure 5C). These outcomes indicated that PRGF administration improves nerve regeneration resulting in better reinnervation of the targeted muscle when compared with only the saline injection.

## 4. Discussion

Following nerve injury, the release of monocyte chemoattractant protein-1 and leukemia inhibitory factors by SCs allows the recruitment of non-polarized macrophages at the injury site [40]. The injury signals trigger the activation and polarization of non-polarized macrophages to the M1 macrophage for phagocytosis of the myelin and axonal debris [40,41]. In the later stage after injury, a switch of macrophages from pro-inflammatory to anti-inflammatory phenotype is essential for the promotion of nerve repair [5]. The failure of the macrophage phenotypic change toward anti-inflammation in IL-10 null mice resulted in the impairment of axonal regeneration and loss of motor and sensory function after nerve crush injury [42]. Other factors in the injured microenvironments may also hinder regeneration. Elevated TNFα prevented phagocytosis-mediated M1 to M2 conversion [40]. The retention of hyperinflammatory factors also prohibits the regenerative capacity of nerves [43]. We observed the extended inflammatory environment and sustenance of M1 macrophages to fail in the regeneration of large peripheral nerve defects [9,10]. On the other hand, the establishment of an immunosuppressive environment and the initiation of M2 macrophages have been shown to control the overstressed immune response [44]. Thus, treatment strategies or drugs for macrophage plasticity and inflammatory modulation are important for nerve regeneration.

Different compositions of PRP and preparation protocols may provide different implications and therapeutic mechanisms. PRP can regulate inflammation and help the transformation of M1 to M2 macrophages depending on the presence or absence of leukocytes [45]. Although both leukocyte-rich (LR) PRP and leukocyte-poor (LP) PRP administration promotes recruitment of M1 macrophages at the injury site initiating early-phase tissue repair in the full-thickness defect model of mice, only LP-PRP induced the activity of M2 macrophages [46]. In the nervous system, anti-inflammatory effects via inhibiting the NFκB pathway were discovered when administrating PRP into a culture of astrocytes [25,47]. Beneficial effects of PRP on nerve regeneration were reported for better functional recovery and an improved sciatic functional index in PRP-treated animals [48]. Filling PRP into the inside-out vein graft as the nerve conduit also increased the axons number, myelinated fibers, and myelin sheath when bridging the sciatic nerve defect over a 10 mm gap [49]. The platelet-rich concentrates have an increased concentration of growth factors, such as transforming growth factor-β (TGF-β), PDGF, and insulin-like growth factor-1 (IGF) that demonstrate positive effects on nerve healing [50,51]. Platelet-rich concentrates also cause macrophage phenotype modification and inhibit inflammation by releasing bioactive molecules (lipoxin) [52]. Platelet-rich fibrin is the second generation of platelet-rich concentrate that has been used to attenuate TNFα, IL-1β, and IL-6 cytokine secretion in LPS-stimulated primary rabbit SCs [53]. This study provided additional evidence for the similar functions of PRGF without the interference of leukocytes on inflammation or nerve regeneration. We also provide the first evidence regarding the potential of PRGF to promote macrophage transition in peripheral nerves by reducing the M1-secreted inflammatory cytokines and number of M1 macrophages (CD68-positive cells) as well as an increase in the expression of M2 macrophage (CD206). Our results agree with other PRGF applications that inhibit the TNFα and IL-1β production by using in vitro rheumatoid arthritis (RA) model [29,54]. The nerve conduit created a relatively close environment after suture and bridged the two ends of an injured nerve. Supplementation of autologous PRP beneficial components without adding more WBC by using the PRGF can be considered to avoid too many immune cells in the conduit.

The lack of blood supply is another bottleneck factor in regeneration [55,56]. SCs migrating on the newly generated vascularized tracks indicate the direct association between angiogenesis and nerve generation [57]. Administration of PRP in the cranial bone defect model resulted in upregulated VEGF and enhancement of angiogenesis [58]. Fibrin present in the PRP acts as a 3D matrix for angiogenesis [59]. The vein graft filled with PRP showed that neo-angiogenesis starts earlier than the sciatic nerve defects treated with nerve autograft [49]. Macrophages are the major non-neuronal cells at the injured site to secrete VEGF-A for angiogenesis and vessel formation. Studies have reflected on the communication between vascular endothelial cells and macrophages through an exchange of contents (microRNA-92a) via extracellular vesicles [60]. Although the PRGF removed the WBC component in PRP, the promotion of angiogenesis was still reported by subcutaneously implanted Matrigel mixed with PRGF to trigger angiogenesis on limb muscles after severe ischemia in mice [61]. In this study, PRGF administration increased the number of RECA-positive endothelial cells indicating a promotion in angiogenesis compared to normal saline. The molecular regulation of angiogenesis and its participating cells through PRGF administration to promote the synchronous events in peripheral nerve regeneration can be investigated in the future.

Taken together, the improvement in healing of a peripheral nerve injury would be unachievable if the focus remained on a single cell and its associated phenomenon rather than targeting multiple stages during repair that can improve the quality of peripheral nerve repair. Our work serves as valuable due to the generation of a regenerative microenvironment through a focus on both macrophages and SCs and laying out the angiogenic network for peripheral nerve regeneration.

## Figures and Tables

**Figure 1 biomedicines-10-01991-f001:**
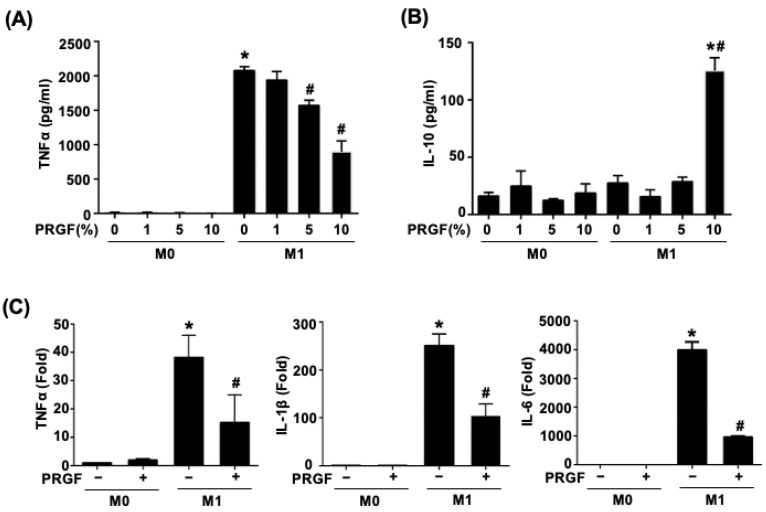
PRGF reduces both the secretion and expression of pro-inflammatory cytokines from M1 macrophages. (**A**) TNFα secretion decreased in M1 macrophages when treated with 5 and 10% PRGF compared to untreated M1 macrophages as checked by ELISA. (**B**) The concentration of anti-inflammatory cytokine IL-10 increased in M1 macrophages after treatment with 10% PRGF for 24 h. (**C**) The increases in mRNA expression levels of TNFα, IL-1β, and IL-6 in M1-induction media were inhibited by adding 10% PRGF as checked by qRT-PCR. n = 3. Significance was assessed by one-way ANOVA. Data are presented as the mean ± SEM. ** p* < 0.05 versus M0. *# p* < 0.05 versus M1.

**Figure 2 biomedicines-10-01991-f002:**
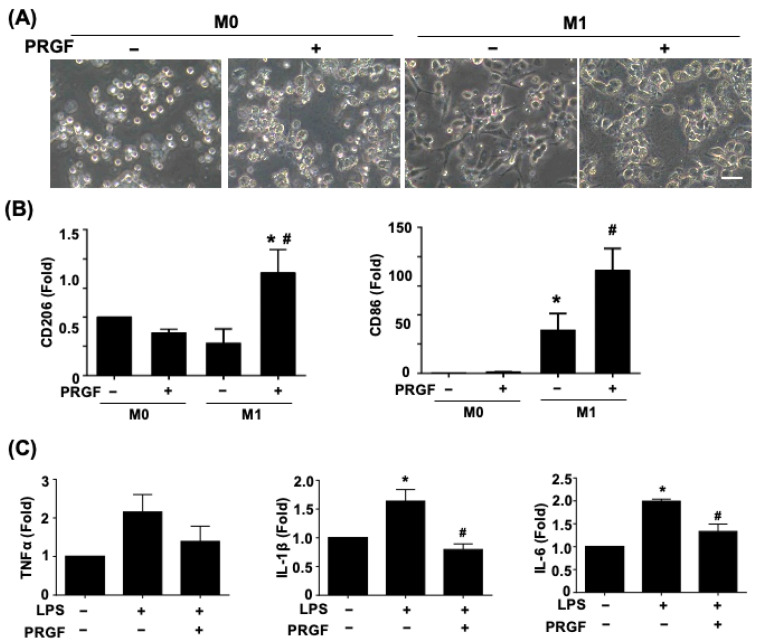
PRGF enhanced M2-macrophage polarization and inhibited the LPS-induced inflammatory SCs. (**A**) Representative phase images revealed a change in cell morphology of PRGF-treated M1 macrophages compared with M1 macrophages. (**B**) The qRT-PCR showed increases in the mRNA expression level for CD206 and CD86 by treating 10% PRGF with the M1-induction media. (**C**) The LPS-induced TNFα, IL-1β, and IL-6 expressions were inhibited when mixing with 10% PRGF in RT4 SCs. n = 4. Significance was assessed by one-way ANOVA. Data are presented as the mean ± SEM. For panel (**B**) histograms; ** p* < 0.05 versus M0. *# p* < 0.05 versus M1. For panel (**C**) histograms; ** p* < 0.05 versus no LPS and PRGF. *# p* < 0.05 versus only LPS. Scale bar = 40 μm.

**Figure 3 biomedicines-10-01991-f003:**
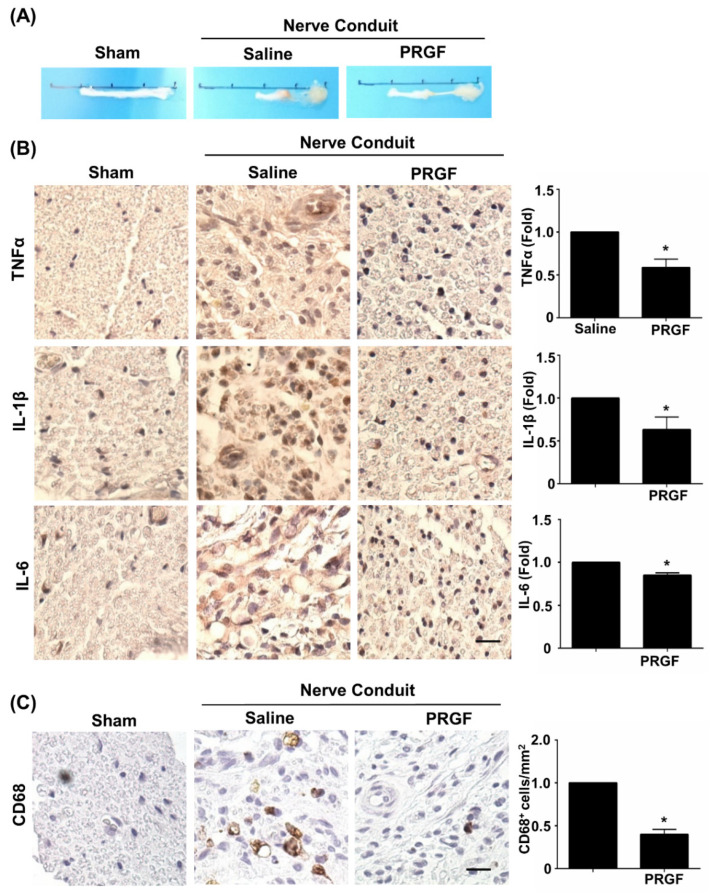
PRGF reduces chronic inflammation and macrophages in nerve conduit. (**A**) Representative pictures of the harvested sciatic nerves after 6 weeks of transection injury and nerve conduit surgery. (**B**) Representative immunohistochemistry (IHC) staining images demonstrated the expression levels of TNFα, IL-1β, and IL-6 in the middle region of transverse sectioned nerve tissue were significantly decreased in the rats that received nerve conduit filled with PRGF. (**C**) The number of M1 macrophages as indicated by the positive marker of CD68 was reduced in PRGF-treated nerve tissue. Scale bar = 20 μm. N = 3. Significance was assessed by paired *t*-test. Data are presented as the mean ± SEM. ** p* < 0.05 versus to saline group.

**Figure 4 biomedicines-10-01991-f004:**
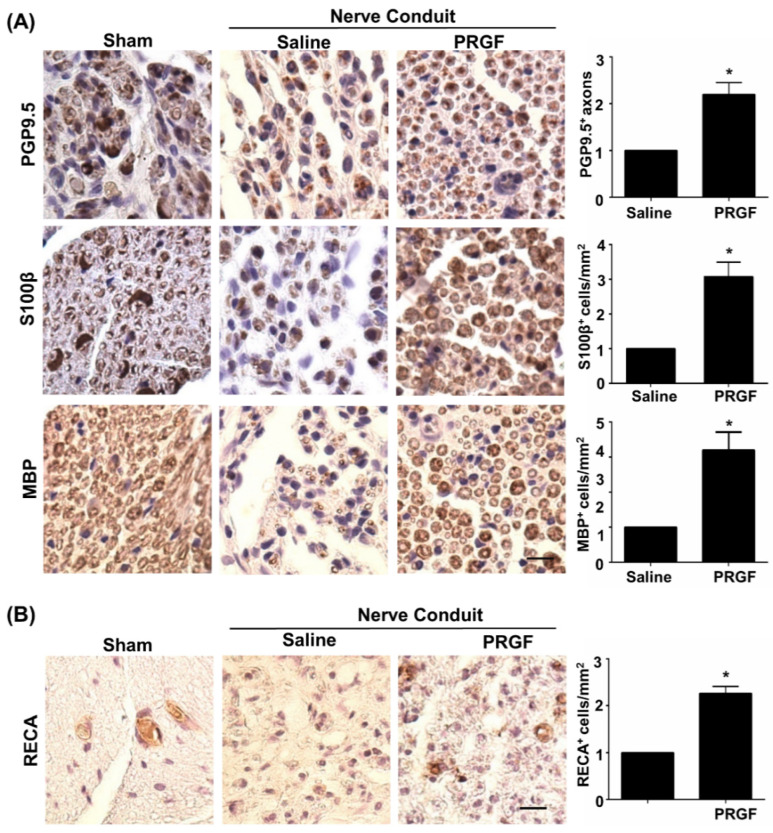
Improvements of nerve regeneration, remyelination, and angiogenesis by PRGF after sciatic nerve injury. (**A**) The IHC staining for PGP9.5, S100β, and MBP demonstrated increases in axon regrowth, SCs numbers, and remyelination after treatment with PRGF for 6 weeks. (**B**) An increase in angiogenesis and vascular formation was observed by IHC staining of rat endothelial cell antigen (RECA). Scale bar = 20 μm. N = 3. Significance was assessed by paired *t*-test. Data are presented as the mean ± SEM. ** p* < 0.05 versus the saline group.

**Figure 5 biomedicines-10-01991-f005:**
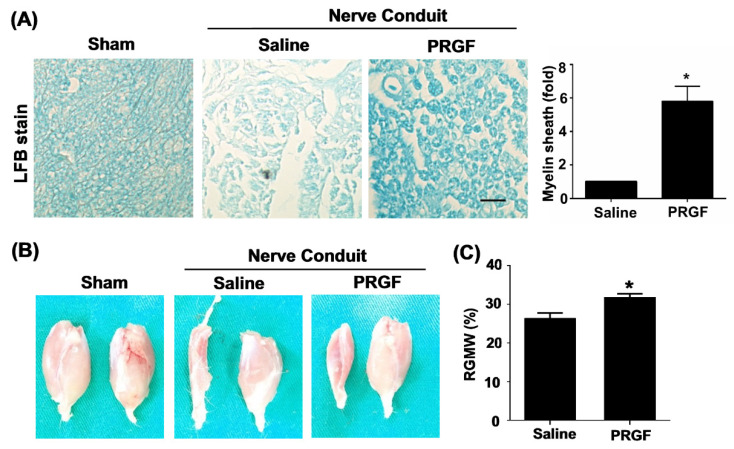
Myelin sheath formation and target muscle innervation confirmed the regenerative outcomes at 6 weeks post-sciatic nerve injury. (**A**) Luxol fast blue (LFB) staining showed an increase in myelin sheath formation for the nerve sections that received PRGF treatment as compared to the saline group. (**B**) The morphology of sciatic nerve innervated target gastrocnemius muscle on an injured (left) and normal (right) leg demonstrated the improvement of regenerated nerve to reinnervate the target muscle for preventing muscle atrophy. (**C**) The relative gastrocnemius muscle weight (RGMW) was quantified and showed an improvement in the PRGF-treated rats. Scale bar = 20 μm. N = 3. Significance was assessed by paired *t*-test. Data are presented as the mean ± SEM. ** p* < 0.05 versus to saline group.

**Table 1 biomedicines-10-01991-t001:** List of Primers for qRT-PCR.

Gene Name	Forward (F)/Reverse (R)	Primer Sequence
TNFα	F	TCAACCTCCTCTCTGCCATC
	R	CCAAAGTAGACCTGCCCAGA
IL1β	F	CTGTCCTGCGTGTTGAAAGA
	R	CTGCTTGAGAGGTGCTGATG
IL6	F	AGGAGACTTGCCTGGTGAAA
	R	CAGGGGTGGTTATTGCATCT
CD86	F	GACGCGGCTTTTATCTTCAC
	R	CCCTCTCCATTGTGTTGGTT
CD206	F	GATGGGTGTCCGAATCTCAG
	R	TTCCACCTGCTCCATAAACC
GAPDH	F	CATCAAGAAGGTGGTGAAGC
	R	TGACAAAGTGGTCGTTGAGG

## Data Availability

The data presented in this study are available in the manuscript.

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
