# Peer review of "Autologous Platelet-Rich Growth Factor Reduces M1 Macrophages and Modulates Inflammatory Microenvironments to Promote Sciatic Nerve Regeneration"

_biomedicines, 2022, doi:10.3390/biomedicines10081991_

Round 1

Reviewer 1 Report

Yadav and colleagues’ study addresses the role of immune modulation in the regenerative outcome of peripheral nerve injuries. They tested the impact of PRGF administration on macrophage polarization both in vitro and in vivo in a rat model of sciatic nerve transection. They found that a mixture of autologous growth factors modulates the inflammatory environment close to the injury favouring the M1 to M2 polarization, thus reducing the release of inflammatory cytokines. This in turn helps the axon to regenerate promptly.

While the topic is relevant for the field of neuroinflammation and neuroregeneration, the description of the experiments sometimes lacks relevant information, and some conclusions drawn by the authors require a more extensive and complete analysis to be confirmed.

Selected criticisms:

-          PRGF was added in % with respect to cell medium. Why not at a defined concentration?

-          Along the same line, how reproducible are the results in vitro using PRGF from different animals? And what about PRGF composition?

-          R4 SC cell line: does SC stand for Schwann cells?

-          Figure 1. I would delete panel A from the figure, as the procedure to purify PRGF is not so innovative as to deserve to be reported in a main figure.

-          Results section 3.1, lines 270-271: ‘The secretion of IL-10 was also increased in M1-induced macrophages (Figure. 1C)’. Do you mean with respect to M0? Apart from the fact that the result is not statistically significant, given that IL-10 is an anti-inflammatory cytokine, why should M1 cells treated with saline only increase IL-10 levels?

-          Again, lines 271-272 ‘The mixture of 10% PRGF in M1 induction medium significantly inhibited the IL-10 secretion’. PRGF increases IL-10 instead!

-          Lines 275-279: ‘The qPCR results showed decreased mRNA expression levels of prototypical M1 cytokines; TNFα, IL-1β, and IL-6 by the treatment of M0 macrophages activated to M1 macrophages with PRGF for 24 h (Figure 1D). The treatment of 10% PRGF in M1 induction medium significantly inhibited the increases of TNFα, IL-1β, and IL-6.’ Repetition of the same concept.

-          RT-PCR experiments lack normalization vs housekeeping gene

-          Is CD206 the only marker of M2 macrophages? And CD68 forM1? I guess whether the different macrophage phenotypes (M1 vs M2) can be identified in a more comprehensive way.

-          Figure 2C. The morphological changes induced by LPS treatment are not clearly appreciable.

-          Figure 2 legend. The sentence related to statistics refers to the histogram in panel B, not that in D.

-          Line 206: Why are you referring to ‘reactive astrocyte morphology’ when describing activated SC?

-          Figure 3: histograms lack * that are described in the legend. Are results statistically significant?

-          What about M2 macrophages in Figure 3? Are they present?

A time course of M1 and M2 staining during the treatment compared to untreated animals is lacking

-          Figure 4. Are data reported in the histograms statistically significant? Longitudinal stainings would be more helpful to visualize and better appreciate the impact of the treatment on vascularization and axonal growth. Moreover, co-localization analysis (via immunofluorescence) would allow to identify the cellule source of the molecules of interest (e.g. cytokines)

-          RECA staining in trasversal sections in Figure 4B is not sufficient to state that the treatment promotes angiogenesis

-          In vivo experiments: the M&M section lacks details on the dose of PRGF employed and the number of administrations.

-          Figure 5, histogram in panel A: statistics? Is the difference statistically significant?

-          Discussion section, lines 405-406. Not clear the following sentence: ‘The PRP can regulate inflammation and help the transformation of M1 to M2 macrophages depending on the composition of the leukocytes’. 

-          No functional evaluation of nerve recovery of function

Author Response

We truly appreciate all the positive comments and suggestions from reviewers. The manuscript is revised according to the reviewer’s comments and improved by the detailed description, and discussions which are indicated by the red color in the new manuscript. The current revision has incorporated the advice and answered the questions point-by-point in red color as below

Reviewer 1

Yadav and colleagues’ study addresses the role of immune modulation in the regenerative outcome of peripheral nerve injuries. They tested the impact of PRGF administration on macrophage polarization both in vitro and in vivo in a rat model of sciatic nerve transection. They found that a mixture of autologous growth factors modulates the inflammatory environment close to the injury favoring the M1 to M2 polarization, thus reducing the release of inflammatory cytokines. This in turn helps the axon to regenerate promptly.

While the topic is relevant to the field of neuroinflammation and neuroregeneration, the description of the experiments sometimes lacks relevant information, and some conclusions drawn by the authors require more extensive and complete analysis to be confirmed.

Selected criticisms:

-          PRGF was added in % with respect to cell medium. Why not at a defined concentration?

    As per the definition of PRGF; it is a complex pool enriched in plasma, platelet-derived proteins, and growth factors. In our study, the focus had been on providing active mediators that may accelerate nerve regeneration rather than one single growth factor. Therefore, we added the PRGF in % with respect to the cell medium instead of a defined concentration. We followed the following paper as a reference [1].

-          Along the same line, how reproducible are the results in vitro using PRGF from different animals? And what about PRGF composition?

    The PRGF preparation used for treatment in each biological replicate (in vitro experiments) was done using different animals. The results were statistically significant indicating the reproducibility in PRGF preparation. Although we have not checked the composition of PRGF in our study, there are other papers that used human or pig blood to prepare PRGF and analyze its composition. PRGF contains cytokines, like IL-1α, IL-1β, IL-2, IL-4, IL-5, IL-6, IL-8, IL-10, TNFα, TNFβ, and IFNγ. Also, there were no significant differences between the concentrations of the cytokines in PRGF obtained from two different species except IL-8 [1,2].

-          R4 SC cell line: does SC stand for Schwann cells?

    We have expanded the acronym SC to Schwann cells in line 197.

-          Figure 1. I would delete panel A from the figure, as the procedure to purify PRGF is not so innovative as to deserve to be reported in the main figure.

       We have deleted panel A from Figure 1.

-          Results section 3.1, lines 270-271: ‘The secretion of IL-10 was also increased in M1-induced macrophages (Figure. 1C)’. Do you mean with respect to M0? Apart from the fact that the result is not statistically significant, given that IL-10 is an anti-inflammatory cytokine, why should M1 cells treated with saline only increase IL-10 levels?

We have used two distinct symbols for showing statistical significance: * means p < 0.05 versus M0, and # means p < 0.05 versus M1 as described in the figure legends. The increase in IL-10 secretion in M1 macrophages treated with 10% PRGF was with respect to M0 and M1. To make the sentence more understandable, we have deleted the earlier lines and changed the statement in lines 313-315.

The secretion levels of IL-10 in M1 macrophages were comparable to M0 levels and were the basal amount with no statistical significance, therefore we considered the increase in IL-10 secretion from M1 macrophages treated with 10% PRGF to be more important. There is literature showing similar levels in their study to check the IL-10 levels in RAW264.7 macrophages differentiated to M1 or M2 macrophages through differential expression of Galectin-9 [3].

-          Again, lines 271-272 ‘The mixture of 10% PRGF in M1 induction medium significantly inhibited the IL-10 secretion. PRGF increases IL-10 instead!

    Apologies for the use of the wrong word. The sentence had to use increased instead of ‘inhibited’. We have corrected it along the previous statement in lines 313-315.

-          Lines 275-279: ‘The qPCR results showed decreased mRNA expression levels of prototypical M1 cytokines; TNFα, IL-1β, and IL-6 by the treatment of M0 macrophages activated to M1 macrophages with PRGF for 24 h (Figure 1D). The treatment of 10% PRGF in M1 induction medium significantly inhibited the increases of TNFα, IL-1β, and IL-6.’ Repetition of the same concept.

    We have deleted the line ‘The treatment of 10% PRGF in M1 induction medium significantly inhibited the increases of TNFα, IL-1β, and IL-6’.

-          RT-PCR experiments lack normalization vs housekeeping gene.

We used ΔΔCt to obtain the normalized relative expression. Briefly, at first, we had normalized the expression values (ΔCt for each condition; the difference between Ct value of gene of interest and housekeeping gene). In the next step, we calculated the difference between ΔΔCt from control (M0) and experimental groups. The last step was to calculate the changes in gene expression levels 2^(ΔΔCt). We stated it in lines 216-221.

-          Is CD206 the only marker of M2 macrophages? And CD68 for M1? I guess whether the different macrophage phenotypes (M1 vs M2) can be identified in a more comprehensive way.

    CD163 and CD206 are reliable markers for M2 macrophages and CD68, CD80, and CD86 for M1 macrophages but at the time of experiment execution we had the accessibility to two available markers in the lab and hence continued to use them.

-       Figure 2C. The morphological changes induced by LPS treatment are not clearly appreciable.

     activated SC?

     We thank you for this suggestion, hence we have removed Figure 2C.

-          Figure 2 legend. The sentence related to statistics refers to the histogram in panel B, not that in D.

    We have added a new sentence describing the statistics for the panel D histograms in the figure 2 legends in lines 362-363.

-          Line 206: Why are you referring to ‘reactive astrocyte morphology’ when describing activated SC?

    We have removed Figure 2C, hence the associated lines have been removed too.

-          Figure 3: histograms lack * that are described in the legend. Are results statistically significant?

    We have checked n=3 times and confirmed the statistical significance. The new figure has been updated.

-          What about M2 macrophages in Figure 3? Are they present?

A time course of M1 and M2 staining during the treatment compared to untreated animals is lacking.

The nerve does not grow within the conduit even up to 4 as we had observed earlier in our lab when we were establishing the sample harvesting time for the sciatic nerve transection injury model. Thus, it was difficult to do the time-course study on a regrown nerve in the conduit region. Hence, for all our studies we continued to perform experiments and analyze the results from nerve tissue harvested at 6 weeks post-injury.

-          Figure 4. Are data reported in the histograms statistically significant? Longitudinal stainings would be more helpful to visualize and better appreciate the impact of the treatment on vascularization and axonal growth. Moreover, co-localization analysis (via immunofluorescence) would allow to identify the cellule source of the molecules of interest (e.g. cytokines)

    The data presented in the histograms is n=3. We have confirmed the statistical significance. The new figure has been updated. The relevant lines have also been added in the figure legends in lines 454-455.

I agree that the vascularization and axonal growth post-treatment would be more evident through the staining of longitudinal sections. Although we tried doing it, the regrown nerve within the conduit is very fine to process for longitudinal sectioning.

-          RECA staining in transverse sections in Figure 4B is not sufficient to state that the treatment promotes angiogenesis

    RECA staining demonstrated the presence of blood vessels. As in Figure 4B, we saw an increase in the number of RECA+ in the PRGF treated group in comparison to the PBS group depicting an increase in the number of blood vessels. Due to this, we stated that the treatment with PRGF promoted angiogenesis.

-          In vivo experiments: the M&M section lacks details on the dose of PRGF employed and the number of administrations.

    For the in vivo experiments, 10% PRGF was used in the conduit as stated in section 2.5. The PRGF was administered only once in the conduit at the time of the surgery. The new sentence has been added in lines 254-255.

-          Figure 5, histogram in panel A: statistics? Is the difference statistically significant?

    We have checked n=3 times and confirmed the statistical significance. The new figure has been updated.

-          Discussion section, lines 405-406. Not clear the following sentence: ‘The PRP can regulate inflammation and help the transformation of M1 to M2 macrophages depending on the composition of the leukocytes’. 

    We would like to further delineate what the sentence meant, thank you for bringing the unclarity to our notice. By ‘composition of the leukocytes’ we meant the presence or absence of leukocytes in PRP. There are studies suggesting the different effects on the distinct stages of tissue repair by administration of either leukocytes-poor (LP) or leukocytes-rich (LR) PRP. Although both leukocyte-rich (LR) PRP and leukocyte-poor (LP) PRP administration promotes recruitment of M1 macrophages at the injury site initiating the early-phase tissue repair in the full-thickness defect model of mice, but only LP-PRP induced the activity of M2 macrophages [4]. Lines 504-509 elucidate the statements.

-          No functional evaluation of nerve recovery of function

    In a previous paper published in our lab with the same sciatic nerve transection model, we had similar outcomes for enhancement in axon regeneration and remyelination, and prevention of effector muscle atrophy when the rats were treated for 6 weeks after surgery. Such improvements resulted in gait recovery in the treated group [5]. We believed an improvement in functional recovery too as PRGF treatment enhanced axon regeneration, remyelination, and prevention of muscle atrophy.

References

  1. Ruzafa, N.; Pereiro, X.; Fonollosa, A.; Araiz, J.; Acera, A.; Vecino, E. Plasma Rich in Growth Factors (PRGF) Increases the Number of Retinal Muller Glia in Culture but Not the Survival of Retinal Neurons. Front Pharmacol 2021, 12, 606275, doi:10.3389/fphar.2021.606275.
  2. Anitua, E.; Prado, R.; Azkargorta, M.; Rodriguez-Suarez, E.; Iloro, I.; Casado-Vela, J.; Elortza, F.; Orive, G. High-throughput proteomic characterization of plasma rich in growth factors (PRGF-Endoret)-derived fibrin clot interactome. J Tissue Eng Regen Med 2015, 9, E1-12, doi:10.1002/term.1721.
  3. Lv, R.; Bao, Q.; Li, Y. Regulation of M1type and M2type macrophage polarization in RAW264.7 cells by Galectin9. Mol Med Rep 2017, 16, 9111-9119, doi:10.3892/mmr.2017.7719.
  4. Nishio, H.; Saita, Y.; Kobayashi, Y.; Takaku, T.; Fukusato, S.; Uchino, S.; Wakayama, T.; Ikeda, H.; Kaneko, K. Platelet-rich plasma promotes recruitment of macrophages in the process of tendon healing. Regen Ther 2020, 14, 262-270, doi:10.1016/j.reth.2020.03.009.
  5. Hsueh, Y.Y.; Chang, Y.J.; Huang, T.C.; Fan, S.C.; Wang, D.H.; Chen, J.J.; Wu, C.C.; Lin, S.C. Functional recoveries of sciatic nerve regeneration by combining chitosan-coated conduit and neurosphere cells induced from adipose-derived stem cells. Biomaterials 2014, 35, 2234-2244, doi:10.1016/j.biomaterials.2013.11.081.

Reviewer 2 Report

The authors report on the effects of PRGF on the interferon g- and LPS-induced expression of pro-inflammatory cytokines TNFα, interleukin-1b, and IL-6 in M1 macrophages. Reductions by PRGF in the levels of mRNAs of these pro-inflammatory cytokines have been shown in in vitro experiments. Moreover, this treatment resulted in the transformation of pro-inflammatory M1 macrophages into the regenerative M2 macrophages.

In in vivo studies the effect of PRGF on nerve regeneration was examined by administering it into the nerve conduit prepared after transection of the sciatic nerve. PRGF improved nerve morphology and increased gastrocnemius muscle mass, suggesting that PRGF treatment may be a new approach to promote functional nerve regeneration.

The study is well designed but there are some points which should be raised.

A significant shortcoming of the in vivo study is the lack of control experiments. Indeed, the effect of PRGF on the intact sciatic nerve was not examined. This is a potentially important issue, since induction of sprouting of intact axons (collateral sprouting) by PRGF may distort the estimation of the numbers of regenerated axons.   

In Fig 4A “PGP9.5cells” probably should read as PGP9.5profiles or axons. 

The photomicrographs in Fig. 3B and C, and Fig 4A,B illustrating the results of sham, saline and PRGF groups, respectively, are apparently not of the same magnification.

line 49: transactional should be replaced with ‘after nerve transection’.

The manuscript would benefit from a linguistic revision of the text.  

Author Response

We truly appreciate all the positive comments and suggestions from reviewers. The manuscript is revised according to the reviewer’s comments and improved by the detailed description, and discussions which are indicated by the red color in the new manuscript. The current revision has incorporated the advice and answered the questions point-by-point in red color as below

Reviewer 2

The authors report on the effects of PRGF on the interferon g- and LPS-induced expression of pro-inflammatory cytokines TNFα, interleukin-1b, and IL-6 in M1 macrophages. Reductions by PRGF in the levels of mRNAs of these pro-inflammatory cytokines have been shown in in vitro experiments. Moreover, this treatment resulted in the transformation of pro-inflammatory M1 macrophages into the regenerative M2 macrophages.

In in vivo studies the effect of PRGF on nerve regeneration was examined by administering it into the nerve conduit prepared after transection of the sciatic nerve. PRGF improved nerve morphology and increased gastrocnemius muscle mass, suggesting that PRGF treatment may be a new approach to promote functional nerve regeneration.

The study is well designed but there are some points which should be raised.

A significant shortcoming of the in vivo study is the lack of control experiments. Indeed, the effect of PRGF on the intact sciatic nerve was not examined. This is a potentially important issue, since induction of sprouting of intact axons (collateral sprouting) by PRGF may distort the estimation of the numbers of regenerated axons.   

We agree with the concerns raised and should have incorporated an uninjured drug-treated sample as well in our study. However, the quantifications done to conclude the improvement of nerve regeneration by PRGF administration in terms of the number of regenerated axons have used untreated-injured (Saline group) for the relative comparison in the treated-injured (PRGF) group.

In Fig 4A “PGP9.5cells” probably should read as PGP9.5profiles or axons. 

The photomicrographs in Fig. 3B and C, and Fig 4A,B illustrating the results of sham, saline and PRGF groups, respectively, are apparently not of the same magnification.

We have corrected the labeling on Y-axis in Fig. 4A from “PGP9.5cells” to PGP9.5+ axons as well as in line 431.

For each marker, the IHC photomicrographs (for each group) were captured at 40X magnification. Although the images have been cropped for representation in the manuscript, the magnification was kept unchanged. Each image in Fig. 3B and C, Fig 4A, and B represent the same magnification, I have double-checked the magnification used to capture the photomicrographs. The concern regarding the appearance of varied nuclei sizes across the groups was even affecting me. The reason behind the varied sizes of nuclei is due to the compactness and non-compactness of the nerve structure. The sham groups have a compact nerve and more cells in comparison to the Saline group, hence the nuclei looked smaller in sham in comparison to the other two.

line 49: transactional should be replaced with ‘after nerve transection’.

   Thank you for the kind reminder. I have replaced transactional with ‘after nerve transection in line 49 and in line 418 in the Figure 3 legends.

The manuscript would benefit from a linguistic revision of the text.  

   We have submitted the revised manuscript to the English editing service as suggested by the journal webpage.

Round 2

Reviewer 1 Report

Auhors' have addressed all criticisms raised by this Reviewer.

Author Response

We thank the reviewer's support. The English editing has been done by journal online service.
